# Can Stringent Government Initiatives Lead to Global Economic Recovery Rapidly during the COVID-19 Epidemic?

**DOI:** 10.3390/ijerph20064993

**Published:** 2023-03-12

**Authors:** Lizheng Ma, Congzhi Zhang, Kai Lisa Lo, Xiangyan Meng

**Affiliations:** 1School of Marxism, East China Normal University, 500 Dongchuan Road, Shanghai 200241, China; 2School of Economics and Management, Shanghai Maritime University, 1550 Haigang Road, Shanghai 201306, China

**Keywords:** COVID-19, government response stringency index, economic recovery

## Abstract

This paper investigates the effectiveness of government measures implemented against COVID-19 and the factors influencing a country’s economic growth from a global perspective. With the help of the data of the Government Response Stringency Index (GRSI), Google mobility, and confirmed COVID-19 daily cases, we conducted a panel model for 105 countries and regions from 11 March 2020 to 31 June 2021 to explore the effects of response policies in different countries against the pandemic. First, the results showed that staying in residential places had the strongest correlation with confirmed cases. Second, in countries with higher government stringency, stay-at-home policies carried out in the early spread of the pandemic had the most effective the impact. In addition, the results have also been strictly robustly analyzed by applying the propensity score matching (PSM) method. Third, after reconstructing a panel data of 47 OECD countries, we further concluded that governments should take stricter restrictive measures in response to COVID-19. Even though it may also cause a shock to the market in the short term, this may not be sustainable. As long as the policy response is justified, it will moderate the negative effect on the economy over time, and finally have a positive effect.

## 1. Introduction

Since the emergence of COVID-19 in December 2019, it spread around the world with alarming speed and caused a series of chain reactions involving aspects of policy, economy, and the environment [1]. It was declared a pandemic by the World Health Organization (WHO) on 11 March 2021. As of 30 June 2022, more than 540 million cumulative cases caused by the virus and over 6.3 million deaths in more than 200 countries had been confirmed, making it the world’s fastest-spreading, most widespread, and most difficult major public health event (PHEIC) [2]. According to statistics released by the World Bank, a severe decline in GDP growth occurred in countries around the world in 2020 and the global economy took a huge hit [3].

However, in addition to the severe economic recession and social crisis, public health security measures have also seriously affected the global response to climate change [4] because the impacts of COVID-19 and climate change easily add up to compound risks [5]. Ebi et al. [6] pointed out that, like the climate crisis, the similarities in the magnitude of the impact, scale, and scope of the response to COVID-19 may pose more severe compounding risks to global socio-economic development, which also provides humanity with a warning about the importance of risk prevention and governance resilience.

In response to the threat caused by this global pandemic, many governments took measures to reduce the increase in the number of confirmed cases and the spread of the epidemic. Governments play an important role in controlling public emergencies. Giné-Garriga et al. [7] examined interventions by governments in various countries in the months before COVID-19; in 84 countries, initiatives were implemented to ensure water, sanitation, and hygiene for all, and authorities sought to provide technical and financial support to service providers. Mawani and Li [8] found that some countries, such as China, Singapore, and New Zealand, did a good job of controlling the outbreak by adopting strict quarantine measures at the beginning of the outbreak, including restrictions on gatherings, transportation closures, and mandatory community quarantines. Wilder-Smith and Freedman [9] thought that the traditional responses to public health measures such as isolation, quarantine, and community isolation played a key role in COVID-19 control before a vaccine was developed.

This article has two objectives. First, we examine whether governments’ intervention measures were effective at mitigating the spread of COVID-19. Second, we explore the impacts of governments’ restriction stringency as regards public social activity on the economic growth of the country. By using weekly OECD macroeconomic panel data for 47 countries from 17 February to 11 September 2020, we attempt to summarize the effects of policy on the epidemic control and economic growth in the early stages of the outbreak of COVID-19.

The rest of the paper is organized as follows. In the Section 2, we provide a literature review. Section 3 introduces the data and the models. Section 4 presents the results of our econometric analysis and robustness tests. Then, we discuss the results of the study in Section 5. Section 6 concludes the paper and makes recommendations.

## 2. Literature Review

Many scholars have examined the relationship between government policies’ strictness and the number of confirmed cases of the outbreak. A study by Yang et al. [10] in 2021 examined the relationship between government restrictions and the number of confirmed new COVID-19 cases and found that the number of confirmed COVID-19 cases and deaths decreased significantly 14–24 days after government restrictions were implemented. Frisina Doetter et al. [11] suggested that state-led policies to address present and/or future public health crises needed to account for the nature of vulnerability in regions. There was also a two-way causal relationship between the number of confirmed cases and strict measures. Asian countries were more concerned with outbreak control and mortality, while non-Asian countries were more concerned with the number of confirmed diagnoses. Using data from 149 countries, Islam et al. [12] revealed that physical distance interventions prior to vaccine development could reduce the number of diagnoses by 13% and that early physical interventions are effective at reducing the incidence of COVID-19.

The impacts of the coronavirus have also been proven a more severe shock for and led to greater uncertainty in countries throughout the world than have past triggers of recession [13,14]. Because the first and most obvious impact of the COVID-19 crisis was the decline in stock markets globally [15], there are also studies focusing on the economic and financial impacts of COVID-19 both globally and regionally [16,17]. Baber and Rao [18] showed that, for India, social distancing policies had a significantly negative impact on economic and business activity and on the stock market, as well as the exchange rate, where economic stimulus had little effect. Hyman et al. [19] used data from information on confirmed cases to successfully predict trends in the number of confirmed cases in the United States and globally, as well as future trends in some leading stock markets. Using daily data for 77 countries from 22 January to 17 April 2020, Ashraf [20] revealed that government quarantine measures led to lower stock market returns and, thus, negatively impacted economic activity, while the positive impact on the economy through a reduction in confirmed cases led to higher returns. König and Winkler [21] exposed the serious negative impact on the economy of COVID-19 and examined the relationship between the rigor index and economic growth in the first and second quarters of 2020.

Existing studies largely affirm that the epidemic has had an extended negative impact on the global economy, especially on the financial and hospitality industries, and the critical role government has played in controlling the spread of the epidemic [22]. However, there is little detailed research on outbreak control, government response stringency, and countries’ economic growth.

According to the International Air Transportation Association (IATA), the air travel industry suffered significant losses due to the COVID-19 outbreak [23]. Wang et al. [24] concluded that the urban population has the strongest correlation with the cumulative number of COVID-19 cases and that cities are the core factor affecting the pandemic. To gain further insight into the study, our research also included data from the Google Community Mobility Report, which reflects the effects of human activities, and the Government Response Stringency Index (GRSI). Google mobility data comprise data that Google presents to measure how effective stringent policies have been at reducing human movement [25]. These data are collected by geographical location in categories of retail and recreation, groceries and pharmacies, parks, transit stations, workplaces, and residential locations, making Google mobility an excellent and effective proxy in the current literature. Unwin et al. [26] confirmed that a reduction in COVID-19 transmission has been explicitly linked to the reduction in mobility in the United States; Vollmer et al. [27] proposed relevant social facilitation measures through the detection of mobility. Nouvellet et al. [28] studied the relationship between the reduction in mobility and COVID-19 transmission. Thus, in this paper, we use Google mobility data to verify the impact of residential mobility status on the number of confirmed outbreak cases by controlling for a strict index of government response. Furthermore, we present evidence that the degree of government policy stringency does have an effect on the country’s economic growth, both long-term and short-term.

## 3. Data and Methods

### 3.1. Government Response Stringency Index (GRSI)

GRSI is a proxy calculated by the Oxford Coronavirus Government Response Tracker (OxCGRT), which tracks comprehensive government interventions through nine standardized indicators [29]. Appendix A explicitly describes the meaning and source of OxCGRT. Many studies have proven that the stringency of these containment measures could potentially reduce the number of confirmed infections [30,31]. In this paper, we first analyze and compare the level of government response stringency with the economic status of countries and the number of confirmed diagnoses during the COVID-19 transmission using GRSI. Second, we collect data on the epidemic from 11 March 2020 to 31 July 2021 for 105 countries and territories. It is worth mentioning that we excluded three countries (the United States, India, and China) from the sample for two reasons. On the one hand, our period of the study coincided with the peak of COVID-19 in the United States [32] and India [33,34], which makes the declared official counts inconsistent with reality and may have an unscientific impact on the results. On the other hand, due to the degree of government disclosure of information, China’s GRSI has a large number of missing values. According to [35], such cases can be defined as a kind of missing completely at random data and we can adopt Listwise Deletion; that is, delete all of the above three individuals with missing values to avoid obvious bias. Finally, we chose the GDP of the 47 countries published on the OECD website for the first quarter of 2020 to the second quarter of 2021 to measure the economic situation of each country after the COVID-19 outbreak.

Figure 1 shows the distribution of GRSI by country globally, and the evolution of COVID-19 can be characterized by four stages. First, according to Figure 1a, most countries’ GRSI levels were at extremely low levels. Second, in the period of the spread of COVID-19, the GRSI of several countries increased to control the spread of the disease. Then, in the third stage, as the global number of new infections rose rapidly, government control efforts began to intensify and GRSI values showed differences in spatial distribution. Most South American countries, such as Chile and Argentina, maintained a high level of government response. Finally, in the middle of 2021, the overall GRSI decreased.

### 3.2. Variables

We first used the logarithm of the weekly number of newly confirmed COVID-19 cases (Confirmed) as one explanatory variable. We selected six variables to measure the change in Google community mobility trends to show how visits and lengths of stay at different places within a geographic area changed compared with the baseline. The baseline was the median value, for the corresponding day of the week, during the 5-week period from 3 January to 6 February 2020. We included mobility trends for places such as: restaurants, cafes, and other retail places (Retail); grocery markets, pharmacies, etc. (Grocery); local parks and some other recreational places (Parks); public transport hubs (Transit); places of work (Workplaces); and for places of residence (Residential). To reduce the volatility of the sample data, we conducted a weekly panel data analysis with the variables Confirmed, Google mobility (Residential, Retail, Grocery, Parks, Transit, and Workplaces), and Stringency.

From a global perspective, on average, trends for places such as local parks, some other recreational places, and residences increased, while several other trends for crowded places decreased. Generally, the public could go out less often, especially to places with dense human traffic during the epidemic, and the tendency to stay at home increased significantly.

To obtain more robust results, we chose more macroeconomic indicators as control variables as follows: the logarithm of GDP (lnGDP), the population (lnpopulation) of the country as control variables, the government expenditure on public health (lnGov_health), the government efficiency index (Gov_effectiveness), the proportion of trade in the GDP (Trade_ratio), and the proportion of the urban population in the total population (Urban_rate) [36,37,38,39]. All of these country-specific variables above allow us to control for the effects that could also have an impact on the Google mobility data related to the number of newly confirmed COVID-19 cases and were obtained from the World Bank [data comes from https://data.worldbank.org, 12 February 2023]. We also added the government policy stringency index (Stringency) to our control variables. Table 1 presents the descriptive statistics of the data.

### 3.3. Models

After processing the data, we achieved a panel data set. The panel data linear regression model was selected for the following advantages. Firstly, compared with cross-sectional models or time series models, the panel data model can control individual heterogeneity better and deal with some unobservable individual effects, which makes the results more convincing. Secondly, it contains more information, which reduces the possibility of collinearity among variables and increases the degree of freedom and the validity of estimation [40]. Thirdly, because of different countries and regions, the causal analysis may contain a set of interfering factors such as economic growth and national crude oil reserves, so, compared to other models, the panel model can estimate the effect of government restrictions on the spread of the pandemic and the economic growth on the premise of individual countries’ characteristics is fixed [41,42]. Appendix B reports related tests for making panel analysis to enhance the understanding of variable trends and the selection of the model.

To verify the effect of the mobility status of the population on the number of confirmed cases of the epidemic, we set up the following model:(1)$$C{\mathrm{onfirmed}}_{it}=\beta_{0}+\beta_{1}Google\_mobility_{it}+\lambda Control_{it}+\mu_{i}+\varepsilon_{it}$$

In model (1), Confirmed*_it_* is the logarithm of the number of newly confirmed daily COVID-19 cases for country *i* at time *t*. *Google_mobility* refers to the six variables indicating the percent changes of Google searches for people’s mobility trends. $\mu_{i}$ is the country fixed effect; $\varepsilon_{it}$ is a random error term.

To analyze the impact of the stringency of the government response on the economies of each country, we assign the following model:(2)$$\ln GDP_{it}=\alpha_{0}+\alpha_{1}Stringency_{it}+\gamma Control_{it}+\eta_{i}+\sigma_{it}$$

In model (2), ln*GDP*
*_it_* is the logarithm of the Gross Domestic Product for country *i* at time *t*. *Stringency_i_*_t_ is the quarterly GRSI data for country *i* at time *t*. $\eta_{i}$ is the country fixed effect; $\sigma_{it}$ is a random error term.

## 4. Results

### 4.1. GRSI, Google Mobility, and the Number of Confirmed Diagnoses

Table 2 displays the regression results of model (1). As can be seen, after controlling for the stringency of the government response, the signs of the coefficients of the corresponding explanatory variables are fully consistent with our expectation that there is a significant negative correlation between the share of stay-at-home persons and the number of newly diagnosed cases. In terms of specific values, the number of confirmed cases decreases by 8.25% for each weekly 1% increase in the share of stay-at-home persons. Similarly, a 1% weekly decrease in visits to workplaces results in a 2.32% decrease in the number of confirmed COVID-19 cases; a 1% weekly decrease in visits to transit stations leads to 4.23% fewer weekly confirmed cases; a 1% weekly decrease in visits to grocery stores and parks results in, respectively, a 2.46% and 1.01% decrease, whereas the effects of visits to retail stores are less influential. Because the changes in mobility trends in the five geographic categories are positively correlated with confirmed cases, except for residential, this suggests that going outside significantly increases the risk of infection. Therefore, we may further infer that a home isolation policy is an effective measure to control the spread of the epidemic. Across all these alternative mobility restriction measures, staying in residential places had the highest impact on reducing confirmed cases, followed by fewer visits to transit stations, grocery stores, workplaces, parks, and retail stores.

According to the descriptive statistics, we find that the median value of Stringency is 60.19. We then discuss the impact of changes in mobility trends on the number of diagnoses by dividing the sample into two groups according to whether or not the Stringency exceeds or is equal to 60.19 (i.e., countries with GRSI more or less than 60.19). The results in Table 3 show that for the group with high Stringency, the stay-at-home share is significantly negatively correlated with confirmed cases at the 1% level. For those with Stringency below 60.19, the coefficient of Residential is nonsignificant. Such situations show that the stricter the government control is, the more effective the impact of stay-at-home orders is, as reflected in the Google mobility data and the number of confirmed diagnoses.

### 4.2. Impact of Government Strictness on Economic Growth

To further examine the relationship between the government stringency index and economic recovery, we conducted another regression using Model (2); the results are described in Table 4. By gradually adding control variables, we find that the stringency coefficient is negatively significant at the 1% level, which can be explained by the fact that for every 1% increase in the government strictness control, the current GDP declines by 0.122%, indicating that the stricter the government control is, the more likely that economic growth will be slow. This result well matched the reality that the global economy suffered a significant negative decline during 2020 and 2021. In terms of work by Silva et al. [43], the U.S. life expectancy at birth dropped by 3.08 years due to the million COVID-19 deaths. Economic welfare losses were estimated to be USD 3.57 trillion. So, it was not surprising to economists that the coronavirus pandemic could plunge the world into a global recession [44]. This may be due to stringent policies, such as the immediate lockdowns of schools and workplaces, international travel restrictions, and public gathering restrictions that the spillover effects of COVID-19 into various markets increased and residents’ economic expectations fell in the short term.

### 4.3. Robustness Tests

In case of a selection bias caused by inter-sample heterogeneity caused by unobserved confounding variables that are constant over time across selected samples [40], many studies have taken the propensity score match (PSM) method to adjust the model [45,46,47]. So, in this paper, we use PSM to verify the positive effect of government control on the reduction in the COVID-19 epidemic. Table 5 shows the results of PSM with those with GRSI exceeding 60.19 as the treatment group and those with GRSI less than 60.19 as the control group. Figure 2 demonstrates the matching effect with the matched K-density map of the treatment group and the control group after PSM. Thus, we conclude that countries with higher GRSI reduce the number of confirmed COVID-19 cases, supporting the previous regression results.

Finally, to further test the effect of GRSI on economic recovery, we lagged the government strictness index by one period (L.Stringency) and examined the effect of current Stringency on economic recovery in the next period. Table 6 reports the results, where the coefficients of the explanatory variables turned positive and passed the 1% significance test. Hence, these findings suggest that the negative effects of government control stringency on the GDP growth of the country are not sustainable. One possible explanation for this may be that stricter government control policies would have a positive impact on future economic recovery by reducing the number of diagnosed cases and, in turn, on future economic recovery [48,49]. In the long term, governments may have enough time to repair their public health care system and other drawbacks in public infrastructure such as the transition to online education, transportation systems, and disease detection systems in public hospitals, as in the UK and Spain [14]. This may also be a good opportunity to fix the country’s economic system and the financial system with a planned federal stimulus package to repair investor sentiments [22].

## 5. Discussion

This paper examines the effects of government policies on controlling the spread of the COVID-19 epidemic, as well as the impact on the country’s economic growth, by using weekly panel data for a sample of major countries around the world. Considering different factors in geography, economy, and society, densely populated metropolitan districts are more likely to reach pandemic status [50]. With the help of Google mobility data and confirmed cases of COVID-19, we managed to measure the effectiveness of people’s responses to control policies. Our findings suggest that reducing social distances and limiting mobility through government interventions can indeed reduce the number of contacts and, thus, the risk of epidemic transmission. This fits well with the previous research and reality status [10,20]. Moreover, we also show that COVID-19 will have a more severe negative impact on the market in the short term, but that this may not be sustainable. If the policy response is justified, it will moderate the negative effect on the economy over time, and finally turn positive.

The results of this study indicate that stay-at-home policies can reduce the spread of COVID-19 to some extent, but only if the government of that country adopts strict control measures. When a country’s government does not adopt strict measures, voluntary isolation of residents and a reduction in leaving the home may not reduce COVID-19 cases. Based on a T-test of the Google Mobility Index, we may preliminarily confirm that compared with countries with relatively loose epidemic prevention policies, residents in countries with more stringent policies paid more attention to the epidemic and had a higher awareness of prevention. This can also explain the phenomenon that the strict measures of isolation at the early stage of the epidemic were effective at preventing the spread of COVID-19 to some extent, which corroborates the reality that many social distancing interventions were proven substantial in response to the COVID-19 pandemic and controlling its transmission both in the Western and Eastern countries by restricting population movement, thus reducing the number of contacts [26,28,51].

We selected GDP data from 47 countries reported by the OECD to analyze the impact of the stringency of the government response on the national GDP and validate it using PSM. Thus, we inferred that the stricter the measures implemented by government, the more likely the country’s economy is to slow down during this period [39,49]. However, we also provided evidence that this is not an inevitable trade-off between human health and a country’s economy in the long-term. A far-sighted stringency policy reduces the virus outbreak and helps maintain a vibrant economy. Until now, with a collaborative effort between policy makers and residents, early isolation, diagnosis, and treatment have proven effective during the pandemic [52] and are expected to help further combat the risk of this infectious disease and recovery as soon as possible after a short period of slow economic growth.

However, this study also has certain limitations that require more research. First, with the changing pattern of COVID-19, uncertainties about the timing and effects of the implementation of government stringent control measures in each country are increasing. This may cause new spatial and temporal heterogeneity of government measures in the post-epidemic era. Because our study only focused on the early period of COVID-19, future research on prospective design is required to further improve the research in this field. Second, the complete isolation of some industries and the shift to online working due to outbreak control measures [53] increased the difficulty of selecting a sample, which may increase the need for future studies with larger samples of different industries from different countries. Third, many institutional and cultural determinants [54] should be taken into account in future research when it comes to policy choices for each country’s policy makers. Even so, this study does not only enrich the literature on the factors used to measure and analyze the impact of stringent policies on the reduction in the spread of COVID-19, but also provides a theoretical perspective to explore the relationship between the effective implementation of government initiatives and the economic recovery of countries.

## 6. Conclusions

As the COVID-19 pandemic spreads around the world, it is not only endangering human lives, but also bringing about severe impacts on public health, the economy, and policy decisions. To alleviate the negative effects of the pandemic, most countries have taken countermeasures. Focusing on the role of the government in this outbreak and the factors influencing a country’s economic growth at the country level, this paper comprehensively discusses the effectiveness of government restriction policies on the control of the COVID-19 pandemic in the early period of the spread.

With the help of data of the Government Response Stringency Index (GRSI), Google mobility confirmed COVID-19 daily cases, we conducted a panel model for 105 countries and regions from 11 March 2020 to 31 June 2021 to explore the effects of response policies in different countries against the pandemic. The main conclusions and relevant policy suggestions of the study are detailed as follows.

First, a fixed-effect panel regression has been employed to show that the government policies of restriction on public mobility had a significantly positive effect on controlling the spread of the pandemic. Among them, staying in residential places had the strongest correlation with the confirmed cases, followed by fewer visits to each of transit stations, grocery stores, workplaces, parks, and retail stores. This indicates that cities having high population mobilities may become a key factor affecting the COVID-19 pandemic. We also infer that control measures, including public transit lockdown, workplace and grocery closures, and gathering restrictions had a positive effect on epidemic control.

Second, we selected the median value of GRSI (60.19) to divide the sample into two groups, further discussing the impact of changes in mobility trends on the number of diagnoses. Our empirical results indicate that the higher the stringency of a country, the more effective will be the impact of stay-at-home policies carried out in the early spread of the pandemic. Furthermore, we robustly analyzed the results by applying the propensity score matching (PSM) method to select countries comparable by dividing groups with whether the GRSI exceeds 60.19 or not. The results showed that the distribution of the control variables for the two groups was almost identical, providing sufficient support to the role of government in limiting the spread of the epidemic.

Third, we also made a further test on the effect of GRSI on economic recovery along time dimensions. By reconstructing panel data of 47 OECD countries, we found that stricter restrictive measures may lead to less slow GDP growth in the early period of the epidemic. However, this may not be sustainable. In the face of COVID-19 public health emergencies, the spread of pandemics can be reduced by timely and strict government measures, such as home quarantines and school and workplace lockdowns, in the absence of pharmaceutical interventions. As long as the policy response is justified, the negative effect on the economy over time will be moderate and eventually turn positive.

Since the imposition of the strict lockdown may have inevitably slowed down the economy, our study may provide evidence that the economic slowdown caused by COVID-19 should be temporary and that it can be reversed to the benefit of economic and health recovery with a long-term series of wise, stringent initiatives. So far, the lockdown policies in most countries have eventually been relaxed so as to restart the economy, corresponding with the conclusions of this paper.

In most cases, it is the national government’s determination, leadership, and ability to make the proper decisions and effectively communicate with the public [24] that determine the outcome of dealing with such emergent crises in a given country. Thus, countries with multiple levels of government must continually work to improve the speed and efficiency of their emergency response systems. In addition, because COVID-19 had a negative impact on certain specifically targeted sectors influenced by this outbreak such as transportation, tourism, groceries, and retail, economic support and subsidies should be required by financial sectors so as to sustain economic revival and prevent a permanent loss caused by both customer behavior and corporate performance. From this perspective, the findings of our study may also provide a good direction for further researchers to focus on the global economic recovery and governments’ decisions, shifting from merely epidemic control to other aspects of public health and economic growth.

## Figures and Tables

**Figure 1 ijerph-20-04993-f001:**
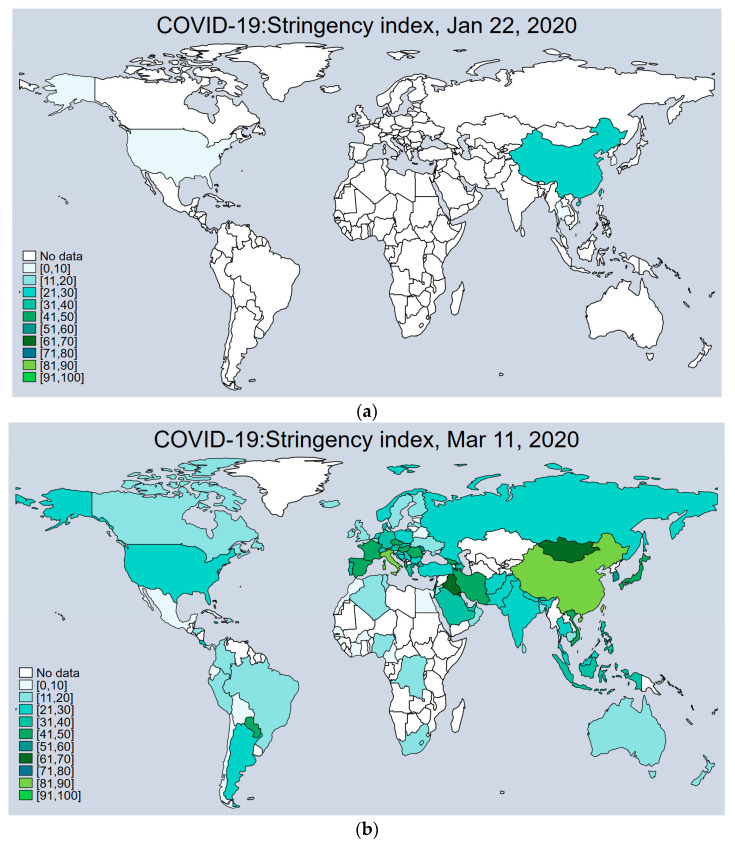
Global GRSI distributions by country separately on (**a**) 22 January 2020; (**b**) 11 March 2020; (**c**) 1 October 2020; (**d**) 25 August 2021. Data Source: Oxford COVID-19 Government Response Tracker, Blavatnik School of Government.

**Figure 2 ijerph-20-04993-f002:**
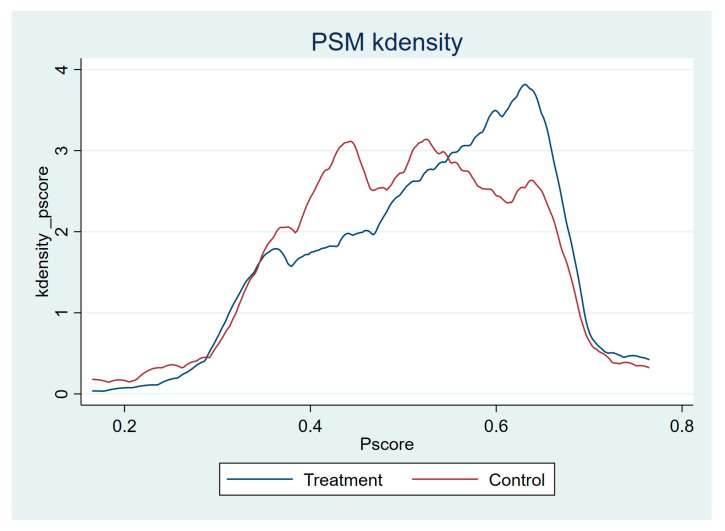
Matched PSM Kdensity map of treatment group and control group.

**Table 1 ijerph-20-04993-t001:** Descriptive Statistics.

Variable	Obs	Mean	Std. Dev.	Min	Max
Confirmed	8165	7.479	2.759	0.000	14.822
Stringency	8165	59.584	18.827	10.714	96.300
Google_mobility					
Residential	8165	7.243	8.413	−11.333	32.06
Retail	8165	−16.935	24.446	−78.361	43.714
Grocery	8165	3.161	24.756	−58.000	86.000
Parks	8165	6.488	47.899	−68.048	177.95
Transit	8165	−21.993	25.648	−77.945	53.952
Workplaces	8165	−20.243	14.598	−62.592	11.565
Control variables					
lnpopulation	8165	16.457	1.508	12.566	21.025
lnGDP	8165	25.490	1.809	21.407	30.696
lnGov_health	8165	3.877	2.208	0.380	9.222
Gov_effectiveness	8165	0.262	0.900	−1.909	2.231
Trade_ratio	8165	91.615	56.538	26.389	381.517
Urban_rate	8165	63.466	21.098	16.350	100.000

Notes: All variables are in weekly frequency, with Confirmed and Stringency as the dependent variables, as well as Google mobility (involving Residential, Retail, Grocery, Parks, Transit, and Workplaces) as the independent variables. The control variables include the population (population, in logarithmic form), the gross domestic product (GDP, in logarithmic form) of the country, government expenditure on public health (Gov_health, in logarithmic form), the government efficiency index (Gov_effectiveness), the proportion of trade in GDP (Trade_ratio), and the proportion of the urban population in the total population (Urban_rate).

**Table 2 ijerph-20-04993-t002:** Impact of Google mobility on the number of new confirmations.

	(1) Confirmed	(2) Confirmed	(3) Confirmed	(4) Confirmed	(5) Confirmed	(6) Confirmed
Residential	−0.0825 ***					
	(−20.51)					
Workplace		0.0232 ***				
		(17.46)				
Transit			0.0423 ***			
			(37.20)			
Retail				0.000635		
				(0.91)		
Grocery					0.0246 ***	
					(17.30)	
Parks						0.0101 ***
						(4.99)
Stringency	0.0528 ***	0.0493 ***	0.0582 ***	0.0279 ***	0.0496 ***	0.0318 ***
	(28.32)	(26.19)	(37.42)	(17.57)	(26.07)	(18.85)
Controls: lnpopulation, lnGDP, Trade_ratio, Gov_effectiveness, Gov_health, Urban_rate						
Constant	−11.14	−27.46	−53.42 *	−55.02	−9.774	−62.62 *
	(−0.34)	(−0.83)	(−1.71)	(−1.61)	(−0.29)	(−1.86)
ID fe	Yes	Yes	Yes	Yes	Yes	Yes
Obs	8165	8165	8165	8165	8165	8165
R^2^	0.5412	0.5349	0.5880	0.5173	0.5346	0.5188

Note: *t* statistics in parentheses; * *p* < 0.10, *** *p* < 0.01.

**Table 3 ijerph-20-04993-t003:** Effect of Google mobility on the number of confirmed diagnoses at different levels of stringency.

Stringency >60.19							Stringency <=60.19					
	Confirmed	Confirmed	Confirmed	Confirmed	Confirmed	Confirmed	Confirmed	Confirmed	Confirmed	Confirmed	Confirmed	Confirmed
Residential	−0.0760 ***						−0.00990					
	(−15.24)						(−1.43)					
Workplace		0.0214 ***						0.00973 ***				
		(12.19)						(4.65)				
Transit			0.0364 ***						0.0323 ***			
			(23.91)						(18.78)			
Retail				−0.00364 ***						0.00119		
				(−3.34)						(1.37)		
Grocery					0.0231 ***						0.0102 ***	
					(12.33)						(4.73)	
Parks						0.0225 ***						−0.0328 ***
						(8.86)						(−10.17)
Constant	279.1 ***	288.8 ***	240.6 ***	295.8 ***	317.3 ***	293.8 ***	5.129	11.25	−9.976	7.713	17.57	25.13
	(3.35)	(3.43)	(3.00)	(3.45)	(3.77)	(3.46)	(0.13)	(0.28)	(−0.26)	(0.19)	(0.43)	(0.63)
Controls: lnpopulation, lnGDP, Trade_ratio, Gov_effectiveness, Gov_health, Urban_rate												
ID fe	Yes	Yes	Yes	Yes	Yes	Yes	Yes	Yes	Yes	Yes	Yes	Yes
Obs	4105	4105	4105	4105	4105	4105	4060	4060	4060	4060	4060	4060
R^2^	0.6318	0.6244	0.6592	0.6115	0.6247	0.6180	0.5581	0.5603	0.5941	0.5581	0.5604	0.5692

Note: *t* statistics in parentheses; *** *p* < 0.01.

**Table 4 ijerph-20-04993-t004:** Short-term economic impact of GRSI.

	(1)	(2)	(3)	(4)	(5)	(6)
	lnGDP	lnGDP	lnGDP	lnGDP	lnGDP	lnGDP
Stringency	−0.00122 ***	−0.00122 ***	−0.00122 ***	−0.00122 ***	−0.00122 ***	−0.00122 ***
	(−6.70)	(−6.70)	(−6.70)	(−6.70)	(−6.70)	(−6.70)
lnpopulation		1.610 ***	1.130 ***	1.190 ***	1.227 ***	0.741 ***
		(123.16)	(60.32)	(69.02)	(71.55)	(13.96)
Gov_effectiveness			0.616 ***	0.698 ***	0.630 ***	0.407 ***
			(25.99)	(25.22)	(22.19)	(17.03)
Gov_health				−0.126 ***	−0.105 ***	0.112 ***
				(−15.35)	(−8.90)	(7.76)
Trade_ratio					0.00163	−0.0126 ***
					(1.32)	(−5.36)
Urban_rate						−0.0835 ***
						(−9.97)
Constant	13.65 ***	−14.70 ***	−6.271 ***	−6.495 ***	−7.329 ***	7.928 ***
	(601.73)	(−59.95)	(−19.10)	(−20.16)	(−18.60)	(4.72)
ID fe	Yes	Yes	Yes	Yes	Yes	Yes
N	294	294	294	294	294	294
R^2^	0.9989	0.9989	0.9989	0.9989	0.9989	0.9989

Note: *t* statistics in parentheses: *** *p* < 0.01.

**Table 5 ijerph-20-04993-t005:** Results of PSM.

Variable Sample	Treated	Controls	Difference	S.E.	T-Stat
lnConfirmed Unmatched	8.22825816	6.72676846	1.50148969	0.058759424	25.55
ATT	8.22825816	6.66344862	1.56480953	0.13051596	11.99
ATU	6.72676846	7.04432736	0.317558898	--	--
ATE	--	0.942635396	--	--	--
Untreated		4073		4073	
Treated		4092		4092	
Total		8165		8165	

Notes: Abbreviation definition: ATT = Average Treatment Effect on the Treated; ATU = Average Treatment Effect on the Untreated; ATE = Average Treatment Effect.

**Table 6 ijerph-20-04993-t006:** Long-term economic recovery impact of GRSI.

	(1)	(2)	(3)	(4)	(5)	(6)
	lnGDP	lnGDP	lnGDP	lnGDP	lnGDP	lnGDP
L.Stringency	0.00214 ***	0.00214 ***	0.00214 ***	0.00214 ***	0.00214 ***	0.00214 ***
	(13.20)	(13.20)	(13.20)	(13.20)	(13.20)	(13.20)
lnpopulation		1.636 ***	1.137 ***	1.204 ***	1.239 ***	0.688 ***
		(135.92)	(65.81)	(75.69)	(78.44)	(14.07)
Gov_effectiveness			0.642 ***	0.732 ***	0.667 ***	0.413 ***
			(29.38)	(28.74)	(25.44)	(18.77)
Gov_health				−0.140 ***	−0.119 ***	0.126 ***
				(−18.50)	(−10.99)	(9.48)
Trade_ratio					0.00157	−0.0146 ***
					(1.38)	(−6.71)
Urban_rate						−0.0946 ***
						(−12.24)
Constant	13.40 ***	−15.41 ***	−6.640 ***	−6.889 ***	−7.693 ***	9.585 ***
	(648.99)	(−68.31)	(−21.93)	(−23.18)	(−21.20)	(6.19)
ID fe	Yes	Yes	Yes	Yes	Yes	Yes
N	252	252	252	252	252	252
R^2^	0.9992	0.9992	0.9992	0.9992	0.9992	0.9992

Note: *t* statistics in parentheses: *** *p* < 0.01.

## Data Availability

Readers can access our data by sending an email to the corresponding author.

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
