# Peer review of "Can Stringent Government Initiatives Lead to Global Economic Recovery Rapidly during the COVID-19 Epidemic?"

_ijerph, 2023, doi:10.3390/ijerph20064993_

Round 1
Reviewer 1 Report (New Reviewer)
I am pleased to read your manuscript. The COVID-19 is a major global public health and environmental problem. It is an important topic for governments to explore the relationship between COVID-19's control measures and economic development in the context of the epidemic. The manuscript uses OLS regression and PSM methods to analyze relevant data, discusses the relationship between the treatment effect of strict government measures on the spread of COVID-19 and the economic recovery of various countries, and draws some valuable conclusions. It is recommended to solve the following problems:
1.In the 1.introduction part, the comprehensive discussion of the existing literature is not deep enough, and the summary of the literature is lacking.
2.The research questions are not connected with the literature review. It is suggested to supplement the problem analysis of the existing literature and thus propose the research issues of this manuscript.
3. The design of some variables in data and methods and the construction of models lack corresponding theoretical support. For example, the causal analysis between personnel mobility and strict government measures still needs to be strengthened. It is suggested to strengthen the theoretical framework analysis of this part, which can refer to the classical theory of infectious disease research.
4. In line 111, "It is worth mentioning that we excluded three countries (the United States, India and China) from the sample." But the data of these three countries appear later. It is suggested to correspond with each other and explain the reasons for excluding the data of three countries.
5.In the part of research results, the discussion of data analysis results is not deep enough, and the discussion part is relatively thin. It is suggested that the author further excavate valuable conclusions, especially the conclusion of 229-235 lines is the highlight of this paper, which is worth more space to discuss.
6. There are some slight mistakes in the manuscript. For example, the format of references, punctuation should be unified:
In Line 12:COVID-2019 weekly data,what is “COVID-2019”?
Some contents in the manuscript are marked yellow. Please pay attention to the uniform format.
7. The spelling of this manuscript requires proofreading and I recommend English language editing by a native speaker.
Author Response
Dear Reviewer,
Thanks a lot for your comments concerning our manuscript entitled Can Stringent Government Initiatives Lead to Global Economic Recovery Rapidly During the COVID-19 Epidemic?(ID: ijerph-2249423) We are glad to respond to your comments and suggestions, and grateful to take this as a great opportunity to improve our manuscript. Thank you for the opportunity to revise our manuscript. We tried our best to revise my paper according to the reviewer’ comments. Besides, we have adopted your kind suggestion to have the English language in our latest version of manuscript proofread. The certificate has also been enclosed for your referrence. On behalf of all authors, I am submitting the revised manuscript for your further assessment. The revised parts are marked highlighted in the paper. The main corrections in the paper and the responses to the reviewer’s comments, point-by-point. Please see the attachment for details.

Reviewer 2 Report (New Reviewer)
The research in this paper has the potential to make a contribution to the literature. However, it is inadequate as presented. I have provided detailed comments in the hopes that you can build on them.
When you write "the stricter the government measures implement, the more likely the country’s economy may slow down during this period". What else was expected? Isn't it too clear without evidence? Can we see the controlled human life losses? I recommend reviewing the following articles for this. I hope you will add it to the review and introduction.
Baber, H. (2020). Spillover effect of COVID19 on the Global Economy. Transnational Marketing Journal (TMJ), 8(2), 177-196.
Baber, H., & Rao, D. T. (2021). THE PRICE OF THE LOCKDOWN-THE EFFECTS OF SOCIAL DISTANCING ON THE INDIAN ECONOMY AND BUSINESS DURING THE COVID-19 PANDEMIC. Ekonomski Horizonti, 23(1), 85-97.
The study provides a good introduction to the study, however, the hook of the study is missing e.g. you are not able to explain to the readers why this study is important and different from the other studies on Covid-19 and the economy.
The title is interesting and suggests to the reader that you may have taken a comprehensive approach yet you may need to reconsider your title in light of the context of the movement or buying behaviour.
There is no Literature review. The literature review needs to be extensive. I don’t see many relevant concepts explained and still not clear to the readers in the introduction. Also, there should be purpose and objective of the research well defined at the end of the section. It is important to highlight why this study is important and how it brings new information to the field of study. The author should provide enough references for the arguments presented in this section.
The review of literature should be more focused on past studies rather than proposing anything new. I don’t see many studies reviewed and referenced in the introduction. It must be reviewed and extended by adding relevant and recent studies. Table 2 is good but the theoretical explanation will add value.
The discussion section is too short for the extensive results presented in the previous section. Readers want to know what these numbers show and propose. I suggest expanding this section and further adding the implications of this research on the practical world.
I wish the author good luck with the revision and look forward to checking the revised version.
Author Response
Dear Reviewer,
Thanks a lot for your comments concerning our manuscript entitled Can Stringent Government Initiatives Lead to Global Economic Recovery Rapidly During the COVID-19 Epidemic?(ID: ijerph-2249423) We are glad to respond to your comments and suggestions, and grateful to take this as a great opportunity to improve our manuscript. Thank you for the opportunity to revise our manuscript. We tried our best to revise my paper according to the reviewer’ comments. Besides, we have the English language in our latest version of manuscript proofread. The certificate has also been enclosed for your referrence. On behalf of all authors, I am submitting the revised manuscript for your further assessment. The revised parts are marked highlighted in the paper. The main corrections in the paper and the responses to the reviewer’s comments, point-by-point. Please see the attachment for details.

Round 2
Reviewer 1 Report (New Reviewer)
The COVID-19 is a major global public health and environmental problem. After the revision of the manuscript, the content of the literature review and discussion is clearer. However, the current manuscript still has the following problems:
1. The narrative logic of the summary does not correspond to the order of the text. It is suggested that the summary can be adjusted.
2. The model adopted in the article is relatively weak in theoretical support.
Author Response
Dear Reviewer,
Thanks a lot for your comments concerning our manuscript entitled Can Stringent Government Initiatives Lead to Global Economic Recovery Rapidly During the COVID-19 Epidemic?(ID: ijerph-2249423) We are glad to respond to your positive and constructive comments and suggestions, and grateful to take this as a great opportunity to improve our manuscript. Thank you for the opportunity to revise our manuscript. We tried our best to revise my paper according to the your comments and have already had the English language in our latest version of manuscript proofread. Please see the attachment.

This manuscript is a resubmission of an earlier submission. The following is a list of the peer review reports and author responses from that submission.
Round 1
Reviewer 1 Report
The Discussion part was omitted from this interesting article. After the presentation of the research results, Conclusions appear immediately, there is a lack of communicative discussion of these results.
The article is interesting and worth publishing. However, in the current version, after presenting the results and statistical descriptions, there is no communicative interpretation of the results for readers who may not be specialists in the field studied.
Typically, such interpretation is given in the chapter titled Discussion, after the Results and before Conclusions. The article lacks this chapter - after the results are given only quite terse conclusions.
Some may have the impression that the article is intended to justify hard-line political restrictions in some countries.
In my opinion, the authors should supplement the article with an objectively formulated Discussion.
Reviewer 2 Report
This paper falls short in methods and technically. It looks like a bad statistical exercise. Moreover, several claims are made by the authors through the text but most of them are unsubstantiated. For example, “countries that implement stringent measures will see a catch-up in economic recovery in the later period due to the lower number of diagnoses ” line 94
There are serious issues regarding the methodology and analysis. Below I name a few.
Google mobility data refer to a very specific population and not the general population. SI is an ancestor of residential mobility. Probably all the effect of SI passes through reducing mobility. Google data measure this mobility—with severe error and not for a representative sample of the population. So what is the point in controlling for SI to estimate the effects of mobility? SI does nothing other than being an instrument for mobility.
Some countries were dropped from the analysis as “outliers”. This is not scientific. The is no discussion about missing data or any rationale for the covariates used which authors refer as “control variables”.
Why bother with Model 1 if this relationship is so well established that authors need to just verify so that they can move to next model. The “Control” matrix does not have subscripts. If these variables are time invariant (and I assume they are different between countries) how did they estimate them with fixed effects? What is their frequency?
Cases are counts but authors chose to model them with a linear model. Moreover, the daily frequency has tremendous volatility (and error). No adjustment was made for weekends, which typically have fewer cases due to underreporting. I recommend aggregating and using Poisson regression.
There is not a word about serial correlation, cross-sectional dependence, unit roots, and why fixed effects were used in the panel model. The dichotomization of SI is arbitrary and stratifying does not have a clear purpose. There are no details on the matching methods either.
Some minor comments.
Abstract. line 16-17. “while in countries with low stringency, increasing the proportion of people at home is not possible ” How is it not possible?
Adding a comma before et al. is not a common citation style.
There is no reason presenting results in the introduction of the paper.
Figure 1 needs serious work. It is not acceptable to present maps in the form a print screens from a website without even removing the margins.
In Table 1 cases have different number of observations the Google data. What does this mean for the analysis?
Please define abbreviations on Table 5.
Round 2
Reviewer 1 Report
Thank you, the manuscript has been sufficiently improved to warrant publication in IJERPH.